# Professional Values and Self-Reported Clinical Competence of Acute Care Nurses in Saudi Arabia: A Cross-Sectional Study

Abdualrahman S. Ashehry [1,*], Ergie P. Inocian [2], Homood A. Alharbi [1], Naif H. Alanazi [1], Norisk M. Adalin [2], Rene P. Carsula [1] and Regie B. Tumala [1]

[1] College of Nursing, King Saud University, Riyadh 12372, Saudi Arabia; homalharbi@ksu.edu.sa (H.A.A.); nalanazz@ksu.edu.sa (N.H.A.); rcarsula@ksu.edu.sa (R.P.C.); rtumala@ksu.edu.sa (R.B.T.)

[2] Nursing Department, King Saud University Medical City, King Saud University, Riyadh 11472, Saudi Arabia; einocian@ksu.edu.sa (E.P.I.); nadalin@ksu.edu.sa (N.M.A.)

[*] Correspondence: abdalshehri@ksu.edu.sa or aalshehry12345@gmail.com

**Abstract:** Background: Professional values are the foundation of nursing practice. Current research evidence on the influence of professional values on clinical competence among acute care nurses in the clinical area is lacking. Purpose: The purpose of this study was to examine the professional values and self-reported clinical competence among acute care nurses. Methods: This quantitative study used a cross-sectional, correlational design. A convenience sample of 403 nurses was recruited to answer the survey utilizing the Nurses Professional Values Scale-3 and the Nurse Competence Scale. The Pearson correlation coefficient was computed to test the relationship between study variables, and a stepwise multiple regression analysis was then performed to investigate the predictors of nurses' professional values and clinical competence. Results: The professional value of "caring" received the highest mean score, followed by "professionalism", while the factor "activism" was rated the lowest. Education was a significant predictor of all three factors of professional values. For clinical competence, "managing situations" was rated as the highest dimension, while "ensuring quality" was rated as the lowest. Moderate positive correlations were revealed between the three factors of professional values and all dimensions of nurses' clinical competence. Area of practice and "activism" were the only significant predictors of the nurses' clinical competence. Conclusions: Nurses perceived all three factors of professional values with high importance in influencing their clinical competence. These findings can contribute to the development of educational interventions to improve and sustain professional values and clinical competence among acute care nurses.

**Keywords:** clinical competence; nurse; professional values; Saudi Arabia

## 1. Introduction

Professional values are considered the foundation of nursing practice [1,2]. Professional nursing values include the nursing principles of human dignity, integrity, altruism, and justice that serve as a framework for standards, professional practice, and evaluation [2]. The American Nurses Association (ANA) Code of Ethics for Nurses with Interpretive Statements delineated nine provisions with the first three focusing on fundamental values and the commitment of nurses [3]. Understanding the professional values of nurses is critically important to providing quality and safe patient care [4,5].

Baccalaureate nursing programs (BSN) promulgate professional nursing values to create a shared professional culture across myriad differences [6]. BSN-prepared nurses provide patient-centered care that identifies, respects, and addresses patients' differences, values, preferences, and expressed needs [7,8]. On the same hand, the Quality and Safety Education for Nurses (QSEN) ensures that nurses are adequately prepared to become competent in clinical practice [9]. The Institute of Medicine developed core competencies in health profession education to include patient-centered care, interdisciplinary teams

(teamwork and collaboration), evidence-based practice, quality improvement, and informatics [8]. These competencies have been adapted by QSEN to evaluate curriculum and course activities for nursing programs. Similarly, the American Association of Colleges of Nursing ensures BSN-prepared nurses are clinically competent to care for patients across the continuum of care in various clinical settings [8]. In fact, the clinical competence of newly graduated nurses was assessed to be at least satisfactory and dependent on prior exposure to the work setting, change in the degree of responsibility, and lack of confidence [10]. Hence, there is a great interest in conducting research on nurses' clinical competence. Various research instruments have been developed and cross-culturally validated to measure nurse competence levels [11–14]. Perhaps the most popular and widely used framework of nursing competence is Benner's novice-to-expert, which details the progression of nurses' ability to perform tasks and achieve desirable outcomes in the clinical context [15]. Acute care nurses in the Kingdom of Saudi Arabia (KSA) are expected to take full responsibility for patients' nursing care in an increasingly complex clinical context. Examination of nurses' clinical competence may assist in both planning and implementing nursing programs, as well as in the development of clinical policy decisions in acute care settings.

Studies on nurses' professional values and clinical competence present diverse findings. For example, a descriptive cross-sectional study that examined the professional values of nurses in Turkey found that nurses had higher scores on the perception of professional values, and education level and years of professional experience were associated with higher scores [16]. A descriptive–analytical study conducted in Iran revealed that nursing students' perspectives toward the professional values' importance were significantly more favorable than those of nurses [1]. A descriptive correlational study of personal and professional factors on job satisfaction and retention indicated a strong correlation between professional values and career development and that both job satisfaction and career development correlated positively with retention [17]. Meanwhile, a cross-sectional study assessing the self-reported clinical competence of newly graduated registered nurses working in Swedish acute care hospital settings found the highest competence in team collaboration and ethics, while the lowest was in professional development and direct clinical practice [18]. In a study comparing the performance levels between hospital nurses and pre-graduate nursing students on the most difficult technical skills, only half of the nurses and one-fifth of the students achieved satisfactory performance levels [19]. A systematic review and meta-analysis of factors related to clinical competence among nurses found that, among individual-related factors, salary has the largest effect size on competence [20]. In another study, length of work experience, age, higher education, permanent employment, and participation in educational programs correlated positively with competence [21]. A mixed-method study in the KSA found the preceptorship program enhanced preceptees' competencies in the clinical setting [22]. Current research evidence on the influence of professional values on the clinical competence of nurses in the clinical area is lacking. It is therefore important to identify factors associated with nurses' professional values and clinical competence to maintain healthcare quality and improve patient outcomes in clinical settings. Thus, this study aimed to examine the professional values and self-reported clinical competence among acute care nurses in the KSA.

## 2. Materials and Methods

### 2.1. Design, Sample, and Setting

A cross-sectional, descriptive study design was used to examine the professional values and self-reported clinical competence among acute care nurses in the KSA. The study was conducted in three (3) tertiary government university hospitals in Riyadh, KSA. During the conduct of the study, the total population of nurses in the study settings was 2000. Using this population size, the sample size was estimated through an online sample size calculator (http://www.raosoft.com/samplesize.html, accessed on 7 November 2020). The computation yielded a sample size of 323 at a 5% margin of error and 95% confidence level. The researchers used convenience sampling. In anticipation of incomplete or unreturned

questionnaires, the informed consent and survey was distributed to 450 clinical nurses. The inclusion criteria for the study were clinical nurses who have been employed for more than a year within the facility and those who voluntarily consented to participate in the study.

### 2.2. Instruments

The current study utilized the Nurses Professional Values Scale (NPVS-3) instrument [5], which reflected the updated American Nurses Association Code of Ethics provision and interpretative statements [3]. It measured the professional values among registered nurses in three factors: caring, activism, and professionalism. The NPVS-3 is a psychometrically validated 28-item 5-point Likert-type scale with response categories ranging from not important (1) to most important (5). Permission to use the tool was obtained from the copyright holder. Internal consistency reliability was reported for the three factors with alpha coefficients ranging from 0.80 to 0.91 and a total scale coefficient of 0.94 [5].

The self-reported clinical competence of the respondents was assessed using the 73-item Nurse Competence Scale (NCS) in a visual analog format corresponding to a response score with a range from 0 to 100 [12]. The total sums in each of the seven competence categories (helping role, teaching–coaching, diagnostic functions, managing situations, therapeutic interventions, and ensuring quality and work role) were calculated. The computed scores were interpreted as low competence (0–25), quite good competence (25–50), good competence (50–75), and very good competence (75–100). A systematic literature review revealed wide use of the scale among experienced nurses, new graduate nurses, and nursing students in hospital settings [21]. The scale also has good concurrent validity and reliability with Cronbach's alpha ranging from 0.79 to 0.9 [12]. Email permission and approval to use the original version of the tool were obtained from the copyright holder.

### 2.3. Ethical Considerations

Ethics approval was sought from the Institutional Review Board of Health Sciences College Research on Human Subjects, the umbrella committee of the three university hospitals (IRB Project No.: E-20-4981). Permission to collect the data was obtained from the nursing administration. Informed consent was presented and obtained to indicate nurses' willful participation in the study. Assurance of confidentiality was upheld throughout the research process.

### 2.4. Data Collection

Data were collected between December 2020 and January 2021. The acute care nurses from each university hospital were visited by the researchers during their working hours. The objectives of the study, significance of the study, risks and benefits, assurance of anonymity, confidentiality of the data, and voluntary participation without any service implications were explained. Clinical nurses in acute care settings were provided ample time, average 30 min., to complete the questionnaire in the staff lounge. There were no rewards of any kind offered to the research participants.

### 2.5. Data Analysis

Data tallying was entered into an Excel spreadsheet and processed using IBM SPSS Version 23.0. The demographic characteristics of the respondents were expressed in frequency counts and percentages, with exception for age, total years of nursing experience, and number of years of nursing experience in the KSA, which were presented in mean and standard deviations. The variables of interest, professional values, and self-reported clinical competence were also analyzed using descriptive statistics such as mean, standard deviations, and range. A stepwise multiple regression analysis was then performed to investigate the relationship between study variables. A decision for statistical significance was taken if the $p$ value was below 0.05.

## 3. Results

### 3.1. Demographic Characteristics of the Respondents

Four hundred and fifty nurses were invited to answer the survey; 403 complete surveys were returned and included in the study, resulting in a response rate of 90%. The average age of the respondents was 38.09 years (SD = 8.78). The majority of the surveyed nurses were females (85.6%), BSN graduates (66.5%), married (63.3%), Christians (80.9%), Filipinos (54.6%), and assigned to inpatient wards (58.6%). The mean total years of experience in the nursing profession was 13.53 (SD = 7.51), while the average number of years of nursing experience in the KSA was 9.97 (SD = 6.99). See Table 1 for the summary of the demographic information of the respondents.

**Table 1.** Demographic characteristics of the respondents (*n* = 403).

| Variable | *n* | % |
|---|---|---|
| Gender | | |
| Female | 345 | 85.6 |
| Male | 58 | 14.4 |
| Education | | |
| Diploma | 120 | 29.8 |
| BSN | 268 | 66.5 |
| Graduate programs | 15 | 3.7 |
| Marital status | | |
| Single | 148 | 36.7 |
| Married | 255 | 63.3 |
| Religion | | |
| Islam | 57 | 14.1 |
| Christian | 326 | 80.9 |
| Hindu | 20 | 5.0 |
| Nationality | | |
| Saudi | 33 | 8.2 |
| Filipino | 220 | 54.6 |
| Indian | 150 | 37.2 |
| Area of practice | | |
| OB-Gyne | 15 | 3.7 |
| Inpatient Wards | 236 | 58.6 |
| OPD | 41 | 10.2 |
| ICU | 39 | 9.7 |
| OR | 28 | 6.9 |
| ER | 44 | 10.9 |
| | Mean (SD) | Range |
| Age | 38.09 (8.78) | 25.00–61.00 |
| Total years of experience as a nurse | 13.53 (7.51) | 1.00–37.00 |
| Years of experience as a nurse in KSA | 9.97 (6.99) | 0.50–33.00 |

### 3.2. Nurses' Perceptions of Nurse Professional Values and Self-Reported Clinical Competence

The findings on the descriptive analyses of the study variables are summarized in Table 2. Among the three factors of the NPV-3, "caring" received the highest mean score of 3.97 (SD = 0.64), followed by "professionalism" (M = 3.79, SD = 0.64). The factor "activism" was rated the lowest factor by the respondents (M = 3.76, SD = 0.65).

For clinical competence, the overall mean score was 90.33 (SD = 12.27). "Managing situations" (M = 91.64, SD = 12.15) was rated as the highest dimension of the nurses' clinical competence, followed by "diagnostic functions" (M = 91.25, SD = 12.38), "work role" (M = 90.21, SD = 12.88), "teaching-coaching" (M = 90.18, SD = 12.51), "therapeutic interventions" (M = 89.88, SD = 13.45), and "helping role" (M = 89.75, SD = 12.71). "Ensuring quality" was rated as the lowest dimension of the clinical competence (M = 89.72, SD = 13.48).

**Table 2.** Mean scores on nurses' professional values and clinical competence ($n$ = 403).

| Variable | Mean | SD | Range |
|---|---|---|---|
| Nurse professional values | | | |
|   Caring | 3.97 | 0.64 | 2.70–5.00 |
|   Activism | 3.76 | 0.65 | 2.20–5.00 |
|   Professionalism | 3.79 | 0.64 | 2.50–5.00 |
| Clinical competence | | | |
|   Helping role | 89.75 | 12.71 | 7.43–100.00 |
|   Teaching–coaching | 90.18 | 12.51 | 6.69–100.00 |
|   Diagnostic functions | 91.25 | 12.38 | 7.00–100.00 |
|   Managing situation | 91.64 | 12.15 | 8.50–100.00 |
|   Therapeutic interventions | 89.88 | 13.45 | 7.10–100.00 |
|   Ensuring quality | 89.72 | 13.48 | 8.17–100.00 |
|   Work role | 90.21 | 12.88 | 7.32–100.00 |
|   Overall clinical competence | 90.33 | 12.27 | 7.33–100.00 |

*3.3. Demographic and Work-Related Predictors of Nurses' Professional Values*

The regression models for "caring" ($F$(16, 386) = 2.74, $p$ < 0.001), "activism" ($F$(16, 386) = 4.16, $p$ < 0.001), and "professionalism" ($F$(16, 386) = 3.55, $p$ < 0.001) were significant. The significant predictors of the three dimensions of NPV-3 are reflected in Table 3. As shown, education was a significant predictor of the three factors, whereas the area of practice was a significant predictor of "activism" and "professionalism". Nationality was also a significant predictor of "activism", while age was a significant predictor of "professionalism".

Specifically, nurses who finished a BSN or a graduate program had significantly higher scores in the three factors of NPV-3 than nurses who finished a diploma in nursing. A year increase in age corresponded to a 0.02-point decrease ($p$ = 0.048) in the "professionalism" mean score. Moreover, Filipino and Indian nurses had significantly lower mean scores in "activism" compared with Saudi nurses. Nurses working in outpatient departments (OPD) reported significantly lower mean scores in "activism" and "professionalism" than inpatient ward nurses, while nurses in an OB-Gyne department had poorer assessments in "professionalism" than inpatient ward nurses.

*3.4. Association between Nurses' Professional Values and Clinical Competence*

The results of the correlation tests between NPVS and clinical competence are depicted in Table 4. Moderate positive correlations were revealed between "caring" and the seven dimensions of clinical competence with Pearson's $r$ ranging from 0.29 to 0.36 ($p$ < 0.001). Similarly, there were moderate positive associations between "activism" and all the dimensions of clinical competence ($r$ range = 0.30–0.38, $p$ < 0.001). Moderate positive relationships were also found between "professionalism" and the seven clinical competence dimensions ($r$ range = 0.30–0.35, $p$ < 0.001).

**Table 3.** Demographic and work-related predictors of nurses' professional values (*n* = 403).

| Predictors | Caring | | | | Activism | | | | Professionalism | | | |
|---|---|---|---|---|---|---|---|---|---|---|---|---|
| | ß | SE-b | *p* | 95% CI | ß | SE-b | *p* | 95% CI | ß | SE-b | *p* | 95% CI |
| Age | −0.01 | 0.01 | 0.193 | −0.03, 0.01 | −0.01 | 0.01 | 0.218 | −0.03, 0.01 | −0.02 | 0.01 | 0.048 * | −0.03, 0.00 |
| Gender | 0.10 | 0.09 | 0.285 | −0.08, 0.28 | 0.11 | 0.09 | 0.234 | −0.07, 0.29 | −0.01 | 0.09 | 0.953 | −0.18, 0.17 |
| Education (Reference: Diploma) | | | | | | | | | | | | |
| BSN | 0.23 | 0.11 | 0.031 * | 0.02, 0.45 | 0.43 | 0.11 | <0.001 *** | 0.22, 0.64 | 0.37 | 0.11 | 0.001 ** | 0.16, 0.58 |
| Graduate programs | 0.65 | 0.20 | 0.001 ** | 0.25, 1.05 | 0.84 | 0.20 | <0.001 *** | 0.45, 1.24 | 0.98 | 0.20 | <0.001 *** | 0.58, 1.38 |
| Marital status | −0.06 | 0.08 | 0.503 | −0.22, 0.11 | −0.04 | 0.08 | 0.673 | −0.20, 0.13 | −0.09 | 0.08 | 0.304 | −0.25, 0.08 |
| Religion (Reference: Islam) | | | | | | | | | | | | |
| Christian | −0.01 | 0.15 | 0.953 | −0.29, 0.28 | −0.21 | 0.14 | 0.154 | −0.49, 0.08 | 0.03 | 0.14 | 0.827 | −0.25, 0.32 |
| Hindu | 0.20 | 0.20 | 0.338 | −0.21, 0.60 | −0.03 | 0.20 | 0.901 | −0.42, 0.37 | 0.10 | 0.20 | 0.635 | −0.30, 0.50 |
| Nationality (Reference: Saudi) | | | | | | | | | | | | |
| Filipino | −0.05 | 0.19 | 0.776 | −0.42, 0.31 | 0.36 | 0.18 | 0.049 * | 0.00, 0.72 | −0.19 | 0.18 | 0.303 | −0.55, 0.17 |
| Indian | −0.15 | 0.18 | 0.409 | −0.51, 0.21 | 0.39 | 0.18 | 0.028 * | 0.04, 0.75 | −0.08 | 0.18 | 0.677 | −0.43, 0.28 |
| Area of practice (Reference: Inpatient wards) | | | | | | | | | | | | |
| OB-Gyne | −0.32 | 0.18 | 0.068 | −0.67, 0.02 | −0.25 | 0.18 | 0.161 | −0.59, 0.10 | −0.70 | 0.18 | <0.001 *** | −1.04, −0.35 |
| OPD | −0.14 | 0.12 | 0.246 | −0.37, 0.09 | −0.44 | 0.12 | <0.001 *** | −0.66, −0.21 | −0.35 | 0.12 | 0.003 ** | −0.58, −0.12 |
| ICU | 0.09 | 0.11 | 0.389 | −0.12, 0.31 | −0.04 | 0.11 | 0.742 | −0.25, 0.18 | −0.02 | 0.11 | 0.891 | −0.23, 0.20 |
| OR | 0.22 | 0.13 | 0.084 | −0.03, 0.46 | 0.03 | 0.12 | 0.803 | −0.21, 0.27 | −0.01 | 0.12 | 0.915 | −0.26, 0.23 |
| ER | 0.14 | 0.11 | 0.188 | −0.07, 0.34 | 0.09 | 0.10 | 0.392 | −0.12, 0.29 | 0.02 | 0.10 | 0.817 | −0.18, 0.23 |
| Total years of experience as a nurse | 0.01 | 0.01 | 0.363 | −0.01, 0.03 | 0.01 | 0.01 | 0.500 | −0.01, 0.03 | 0.01 | 0.01 | 0.164 | −0.01, 0.03 |
| Years of experience as a nurse in KSA | 0.01 | 0.01 | 0.412 | −0.01, 0.03 | 0.01 | 0.01 | 0.175 | −0.01, 0.03 | 0.01 | 0.01 | 0.127 | −0.00, 0.03 |
| $R^2$ (Adjusted $R^2$) | 0.102 (0.065) | | | | 0.147 (0.112) | | | | 0.128 (0.092) | | | |

Note. Caring, activism, and professionalism were the dependent variables. β is the unstandardized coefficient; SE-b is the standard error. * Significant at 0.05, ** significant at 0.01, *** significant at 0.001.

**Table 4.** Association between nurses' professional values and clinical competence (*n* = 403).

| | Caring | | Activism | | Professionalism | |
|---|---|---|---|---|---|---|
| | *r* | *p* | *r* | *p* | *r* | *p* |
| Helping role | 0.36 | <0.001 *** | 0.38 | <0.001 *** | 0.35 | <0.001 *** |
| Teaching–coaching | 0.34 | <0.001 *** | 0.34 | <0.001 *** | 0.33 | <0.001 *** |
| Diagnostic functions | 0.33 | <0.001 *** | 0.34 | <0.001 *** | 0.33 | <0.001 *** |
| Managing situation | 0.31 | <0.001 *** | 0.30 | <0.001 *** | 0.30 | <0.001 *** |
| Therapeutic interventions | 0.29 | <0.001 *** | 0.31 | <0.001 *** | 0.31 | <0.001 *** |
| Ensuring quality | 0.33 | <0.001 *** | 0.35 | <0.001 *** | 0.35 | <0.001 *** |
| Work role | 0.33 | <0.001 *** | 0.34 | <0.001 *** | 0.33 | <0.001 *** |
| Overall clinical competence | 0.34 | <0.001 *** | 0.35 | <0.001 *** | 0.34 | <0.001 *** |

Note. *** Significant at 0.001 level.

### 3.5. Demographic and Work-Related Predictors of Nurses' Clinical Competence

The three factors of NPVS-3, as well as the demographic and work-related variables, were entered into a multiple regression analysis to predict the overall clinical competence of the nurses. The regression model was significant ($F_{(19, 383)}$ = 4.41, $p$ < 0.001), predicting approximately 13.9% ($R^2$ = 0.179, Adjusted $R^2$ = 0.139) of the variance of the nurses' overall clinical competence. As shown in Table 5, the area of practice and "activism" were the only significant predictors of the nurses' clinical competence. Nurses in an OB-Gyne department reported significantly lower clinical competence than nurses in the inpatient ward. Moreover, a point increase in the mean score of "activism" resulted in a 4.75-point ($p$ = 0.041, 95% CI = 0.19, 9.32) increase in the overall clinical competence mean score.

**Table 5.** Demographic and work-related predictors of nurses' clinical competence (*n* = 403).

| Predictors | ß | SE-b | *p* | 95% CI | |
|---|---|---|---|---|---|
| | | | | Lower | Upper |
| Age | 0.02 | 0.16 | 0.899 | −0.30 | 0.34 |
| Gender | −1.41 | 1.71 | 0.408 | −4.76 | 1.94 |
| Education (Reference: Diploma) | | | | | |
| BSN | 2.28 | 2.06 | 0.268 | −1.77 | 6.33 |
| Graduate programs | 6.86 | 3.90 | 0.080 | −0.82 | 14.53 |
| Marital status | −0.54 | 1.54 | 0.724 | −3.56 | 2.48 |
| Religion (Reference: Islam) | | | | | |
| Christian | 0.68 | 2.73 | 0.803 | −4.68 | 6.05 |
| Hindu | −2.66 | 3.79 | 0.483 | −10.10 | 4.78 |
| Nationality (Reference: Saudi) | | | | | |
| Filipino | −2.53 | 3.61 | 0.483 | −9.62 | 4.56 |
| Indian | 2.85 | 3.54 | 0.421 | −4.12 | 9.82 |
| Area of practice (Reference: Inpatient wards) | | | | | |
| OB-Gyne | −8.97 | 3.43 | 0.009 ** | −15.71 | −2.24 |
| OPD | 0.95 | 2.26 | 0.676 | −3.49 | 5.38 |
| ICU | 1.18 | 2.04 | 0.565 | −2.84 | 5.19 |
| OR | −2.82 | 2.35 | 0.232 | −7.44 | 1.81 |
| ER | −1.91 | 1.95 | 0.328 | −5.74 | 1.92 |
| Total years of experience as a nurse | −0.09 | 0.18 | 0.629 | −0.45 | 0.27 |
| Years of experience as a nurse in KSA | 0.26 | 0.17 | 0.138 | −0.08 | 0.59 |
| Caring | 3.94 | 2.18 | 0.071 | −0.34 | 8.22 |
| Activism | 4.75 | 2.32 | 0.041 * | 0.19 | 9.32 |
| Professionalism | −1.26 | 2.31 | 0.586 | −5.79 | 3.28 |

Note. Clinical competence was the dependent variable; β is the unstandardized coefficient; SE-b is the standard error. $R^2$ = 0.179; adjusted $R^2$ = 0.139. * Significant at 0.05, ** significant at 0.01.

## 4. Discussion

This study investigated the professional values and self-reported nursing competence among acute care nurses in the KSA and their associated factors. Critical findings on nurses'

perceptions of their professional values and clinical competence were obtained. These findings can contribute to the development of educational interventions for sustaining and enhancing the professional values and clinical competence of nurses.

The findings revealed that nurses rated the three factors of the NPVS-3 with high importance. "Caring" received the highest mean score, followed by "professionalism", while the factor "activism" was rated the lowest. Generally, the finding is comparable to the results of other studies conducted in various countries [1,23,24]. Nurses perceived themselves highly committed to caring for individual patients and their families. They also feel and act responsibly toward their working environment, personal and professional growth, and clinical practice. However, nurses put their dynamic roles in the profession last. This value is crucial to sustain nursing standards, uphold health diplomacy, and impact health policy. Nursing leaders and educators can use this information to tailor interventions to improve the professional values among nurses. For example, a structured professional values development program showed promising results [25].

The regression models of the demographic and work-related variables were found to be significant predictors of nurses' professional values. However, the literature suggests that the contradicting information of the socio-demographic and professional characteristics primarily influences the professional caring behavior among nurses [26], calling for the need for further investigation to fully understand the variations in nurses' professional values. Specifically, the study found that nurses who finished a BSN or a graduate program had significantly higher scores in the three factors of NPVS-3 than nurses who finished a diploma in nursing. It is therefore important to reiterate professional values in higher education programs.

Additionally, nurses working in OPD reported significantly lower mean scores in "activism" and "professionalism" than inpatient ward nurses, while nurses in an Ob-Gyne department had poorer perceptions of "professionalism" than inpatient ward nurses. This lower level of perceptions could be explained by the routine-based practice of OPD and Ob-Gyne nurses in their units, which may have affected their professional values. This further implies that, perhaps, the respondents had set aside or neglected patient-centered care needs as upheld by the American Association of Colleges of Nursing [7] and Institute of Medicine [8] and, instead, imitated the professional behaviors and attitudes of their colleagues who worked for many years in the unit following routine-based practice. Routine-based practice has been reported as one of the barriers of developing clinical competence among master's level students and graduates of gerontological nursing in an Iranian qualitative study [27] and routine-based education as an inhibitory trait of clinical instructors for developing clinical competence among nursing students in another qualitative study in Iran [28]. However, the findings and explanations presented above must be interpreted with caution as there could be other explanations (e.g., lack of confidence, poor self-esteem, job dissatisfaction, cultural aspect, type of nursing leadership in the unit, and others) as to why OPD nurses had lower levels of perceptions on "activism" and "professionalism" and Ob-Gyne nurses on "professionalism" compared to inpatient ward nurses in the current study. Hence, nurse educators can formulate effective educational strategies to professional values such as value-based education that could influence moral character and provide significant contributions to the attainment of excellent nursing practice [29,30]. Surprisingly, age had a negative correlation with "professionalism", wherein a year increase in age corresponded to a decrease in the "professionalism" mean score. This particular finding contradicted the results of a former study among Saudi nursing students, wherein age significantly and positively predicted their perceived professionalism values [31]. Traditionally, nurses gain more experience as they become older, which could facilitate their personal and professional transformation leading to enhancement of their professional values and self-efficacy to care for patients as reflected in their greater awareness of skills and performance [1,23]. The result of the current study might have been affected by the diverse nature of acute care nurses working in the KSA, where the majority of the workforce are expatriate nurses from India and the Philippines. Since expatriate

workers cannot be granted citizenship, many are transitioning to Western countries (i.e., Canada, the UK, the USA, Australia, or New Zealand) [32]. The nationality of nurses was also found to be a significant predictor of their professional values on the "activism" factor. The findings showed that Filipino and Indian nurses had significantly lower mean scores in "activism" compared with Saudi nurses. Expatriate nurses might feel inferior, which leads to having limited autonomy and a feeling of powerlessness to exercise patient advocacy [33,34].

For clinical competence, "managing situations" was rated as the highest dimension, while "ensuring quality" was rated as the lowest. The results affirmed the important roles of acute care nurses to ensure optimal health outcomes. However, multiple methods may be required to obtain comprehensive and accurate assessments of nurses' competence in acute care settings [35]. The findings showed that there is room for improvement in ensuring that nurses become fully competent in their work role. Acute care nurses must engage in evidence-informed decision-making, also known as evidence-based practice [36]. The clinical decision support system (CDSS) demonstrated provided tangible support and necessary resources to improve the diagnostic accuracy of nurses in practice [37]. Hyun et al. [38] found a strong disconnect between the competency level of new graduates and work readiness. New graduates were well prepared for demonstrating respect to patients but needed to be closely supported when providing emergency care. Mentoring can assist the nurses to transition to their new roles and develop knowledge and skills to advance practice [22,39]. This is also supported by McNamara et al.'s [40] findings that mentoring, coaching, and action learning were positively experienced by participants and contributed to the development of clinical leadership competencies. In addition, Naef et al. [41] recommended the adoption of primary nursing to improve quality of care in acute care inpatient settings and develop patient-centered caring.

The results affirmed the influence of professional values on clinical competence among nurses. The results were consistent with another study in the KSA, which reported the interrelationship of nurses' professional values and clinical competence through the use of a structural equation-modeling approach [42] The humanistic nature of nursing practice cultivates authentic relationships and altruistic behaviors for the greater good of the patients, which constitutes the cornerstone of the profession [26,43,44]. The application of professional values where nurses engage in active participation in nursing activities using evidence-based practice may help achieve overall clinical practice competencies [24]. The active participation of nurses must be evident in the following aspects: advocacy for equal opportunities to the provision of nursing care, involvement in professional organizations, multidisciplinary collaboration, public policy decision-making, recognition of the healthcare needs of patients, research work and scientific publication, and standardization of nursing practice [24].

This study has identified a few limitations. Careful interpretation of the findings should be considered. The self-reported collection of responses from the study respondents may have a degree of social desirability bias. Sampling bias might have also occurred given the non-random selection of the respondents. Furthermore, the cross-sectional nature of this study hindered the examination of causal relationships between variables. It is recommended to conduct longitudinal studies that will capture the in-depth nature of the nurses' professional values and clinical competence. Also, the current study environments might limit the generalizability of the results to other Arab countries and internationally, although the study was conducted on a larger scale comprising multiple sites in the KSA. Lastly, another significant limitation pertains to the study's focus on associations between self-reported data rather than direct observations and views and opinions of acute care nurses; hence, future qualitative studies are warranted.

## 5. Conclusions and Recommendations

Our study concluded that the acute care nurses in the KSA perceived the three professional values of "caring", "activism", and "professionalism" as having high importance

in influencing their clinical competence. However, "activism", which is critical to sustain nursing standards, uphold health diplomacy, and impact health policy, was rated lower than the other two values. Among the dimensions of nurses' clinical competence, "ensuring quality" was rated the lowest, which is essential to ensure safe patient care. For nursing management and education, the findings implied that there are some areas that should be focused on, such as the significant correlations between the demographic characteristics, professional values, and self-reported clinical competence of acute care nurses. For example, education was a significant predictor of all three factors of professional values. Moderate positive correlations were revealed between the three factors of professional values and all dimensions of nurses' clinical competence. Lastly, the area of practice and the professional value of "activism" were the only significant predictors of the nurses' clinical competence.

As improvements in professional values may positively impact the clinical competence among acute care nurses, we recommend that the findings of our study be utilized by nursing leaders and educators to develop effective educational interventions that improve and maintain the professional values and clinical competence among nurses to ensure safe and quality patient care. For example, a structured educational program can be formulated geared to developing moral character among acute care nurses. In addition, nursing leaders and educators should encourage active participation among acute care nurses in quality improvement activities and evidence-based practice. Further studies should be conducted to capture the in-depth nature of the nurses' professional values and enhance clinical competence. Efforts should be undertaken to fully integrate all professional nursing values in the clinical context in the KSA and other Arab countries.

**Author Contributions:** Conceptualization, A.S.A. and E.P.I.; methodology, H.A.A., R.P.C. and N.M.A.; software, validation, and formal analysis, A.S.A., N.H.A. and R.B.T.; investigation, E.P.I., R.B.T. and N.M.A.; resources, A.S.A.; data curation, A.S.A., N.H.A. and R.B.T.; writing—original draft preparation, H.A.A., R.P.C. and E.P.I.; writing—review and editing, A.S.A., R.B.T. and N.M.A.; visualization, A.S.A. and N.H.A.; supervision, H.A.A.; project administration, A.S.A. and N.M.A.; and funding acquisition, A.S.A. and H.A.A. All authors have read and agreed to the published version of the manuscript.

**Funding:** The researchers are thankful to the Researchers Supporting Projects with grant number (RSPD2023R879), King Saud University, Riyadh, Saudi Arabia for supporting this research project.

**Institutional Review Board Statement:** The study was conducted in accordance with the Declaration of Helsinki and approved by the Institutional Review Board of King Saud University Medical City (protocol code: E-20-4981 on 15 November 2020).

**Informed Consent Statement:** Informed consent was obtained from all respondents in this study.

**Data Availability Statement:** The data presented in this study are available on request from the corresponding author.

**Conflicts of Interest:** The authors declare no conflict of interest.

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
