# Peer review of "Professional Values and Self-Reported Clinical Competence of Acute Care Nurses in Saudi Arabia: A Cross-Sectional Study"

_ejihpe, doi:10.3390/ejihpe13110186_

Round 1

Reviewer 1 Report

Comments and Suggestions for Authors

Thank you for the study.

The paper is clearly written. 

Perhaps the conclusion should provide a succinct integration of your study and not repeat/ provide new points.

Good to provide some recommendations from your study to add value to the study.

Author Response

Thank you for the study. The paper is clearly written. 

REPONSE: Thank you very much for your positive review of our paper. We revised our work based on your valuable comments and provided our point-by-point response to each comment.

Perhaps the conclusion should provide a succinct integration of your study and not repeat/ provide new points.

REPONSE: We revised 'Section 5. Conclusions' as suggested. Thank you.

Good to provide some recommendations from your study to add value to the study.

REPONSE: We renamed ‘Section 5. Conclusions’ to ‘Section 5. Conclusions and Recommendations’ and provided additional recommendations at the second paragraph of this section. Thank you.

Reviewer 2 Report

Comments and Suggestions for Authors

The study was conducted appropriately. The writing is at the appropriate scholarly level.

The only suggestion I have is to consider adding qualitative studies to understand the differences in 'areas of practice' of the regression predictors. Understanding why Ob-Gyn was significant compared to other areas might be enlightening.

Author Response

The study was conducted appropriately. The writing is at the appropriate scholarly level.

REPONSE: Thank you very much for your positive review of our paper. We revised our work based on your valuable suggestion below and provided point-by-point response. 

The only suggestion I have is to consider adding qualitative studies to understand the differences in 'areas of practice' of the regression predictors. Understanding why Ob-Gyn was significant compared to other areas might be enlightening.

REPONSE: We added two qualitative studies to possibly explain this finding and updated the references accordingly. If you require further explanations/revisions, we are very willing to comply. Thank you.

Reviewer 3 Report

Comments and Suggestions for Authors

1.Kindly format the paper as per journal guidelines See examples of published papers for headings, labels for figures, and tables (should be self-explanatory).

2. Update the references to reflect the latest in the field and compare the existing studies with your results in detail.

3. Expand on limitations and strengths section. 

4. Elaborate on implications of your work (for practice, research, prevention, policy, etc.- whichever is applicable)

5. Please check for errors with syntax, sentence composition, grammar, spelling, punctuations, etc.

6. The abstract should have greater emphasis on results and methods. Please edit accordingly.

Comments on the Quality of English Language

quality is ok

Author Response

1.Kindly format the paper as per journal guidelines See examples of published papers for headings, labels for figures, and tables (should be self-explanatory).
REPONSE: Thank you very much for your positive review of our paper. We revised our work based on your valuable comments and provided point-by-point response to each comment. For this comment, we revised the headings and labels for tables as suggested, especially in the Results section of the study.

2. Update the references to reflect the latest in the field and compare the existing studies with your results in detail.

REPONSE: We added the citations (qualitative studies) to support the explanations of the findings in the Discussion section as also suggested by Reviewer2, and updated the references accordingly.

3. Expand on limitations and strengths section. 

REPONSE: This part has been revised and expanded as suggested.

4. Elaborate on implications of your work (for practice, research, prevention, policy, etc.- whichever is applicable)

REPONSE: We revised this part of our study in the Conclusions and Recommendations section.

5. Please check for errors with syntax, sentence composition, grammar, spelling, punctuations, etc.

REPONSE: We have checked this and if the honorable reviewer would suggest further English editing of our work, we are very willing to comply.

6. The abstract should have greater emphasis on results and methods. Please edit accordingly.

REPONSE: We revised and updated the Abstract part of this study as suggested.

Round 2

Reviewer 3 Report

Comments and Suggestions for Authors

Thanks for all the revisions

Comments on the Quality of English Language

May benefit from minor editing